# Single Nucleotide and Copy-Number Variants in IL4 and IL13 Are Not Associated with Asthma Susceptibility or Inflammatory Markers: A Case-Control Study in a Mexican-Mestizo Population

**DOI:** 10.3390/diagnostics10050273

**Published:** 2020-04-30

**Authors:** Enrique Ambrocio-Ortiz, Gustavo Galicia-Negrete, Gloria Pérez-Rubio, Areli J. Escobar-Morales, Edgar Abarca-Rojano, Alma D. Del Angel-Pablo, Manuel D. J. Castillejos-López, Ramcés Falfán-Valencia

**Affiliations:** 1HLA Laboratory, Instituto Nacional de Enfermedades Respiratorias Ismael Cosío Villegas, Calzada de Tlalpan 4502, Sección XVI, Mexico City 14080, Mexico; e_ambrocio_iner@hotmail.com (E.A.-O.); dr.gustavogalicia@gmail.com (G.G.-N.); glofos@yahoo.com.mx (G.P.-R.); arelijoaline@hotmail.com (A.J.E.-M.); alyde_08@hotmail.com (A.D.D.A.-P.); 2Sección de Estudios de Posgrado e Investigación, Escuela Superior de Medicina, Instituto Politécnico Nacional, Plan de San Luis y Díaz Mirón s/n, Casco de Santo Tomas, Mexico City 11340, Mexico; rojanoe@yahoo.com; 3Epidemiological Surveillance Unit, Instituto Nacional de Enfermedades Respiratorias Ismael Cosío Villegas, Calzada de Tlalpan 4502, Sección XVI, Mexico City 14080, Mexico; mcastillejos@gmail.com

**Keywords:** asthma, SNP, CNV, *IL4*, *IL13*, IgE

## Abstract

Background: Asthma is a complex and chronic inflammatory airway disease. Asthma’s etiology is unknown; however, genetic and environmental factors could affect disease susceptibility. We designed a case-control study aimed to evaluate the role of single-nucleotide polymorphisms (SNP), and copy-number variants (CNV) in the *IL4* and *IL13* genes in asthma susceptibility and their participation in plasma cytokine levels depending on genotypes Methods: We include 486 subjects, divided into asthma patients (AP, *n* = 141) and clinically healthy subjects (CHS, *n* = 345). We genotyped three SNP, two in the *IL4* and two in the *IL13* gene; also, two CNVs in *IL4*. The IL-4, IL-13 and IgE plasma levels were quantified. Results: Biomass-burning smoke exposure was higher in the AP group compared to CHS (47.5% vs. 20.9%; *p* < 0.01, OR = 3.4). No statistical differences were found in the genetic association analysis. In both CNV, we only found the common allele. For the analysis of IL-4, IL-13, and IgE measures stratified by genotypes, no significant association or correlation was found. Conclusion: In the Mexican-mestizo population, SNPs neither CNVs in *IL4* nor *IL13* are associated with asthma susceptibility or involved serum cytokine levels. Biomass-burning smoke is a risk factor in asthma susceptibility.

## 1. Introduction

Asthma is a chronic inflammatory disease characterized by airway hyperresponsiveness, causing recurrent dyspnea, cough, and wheezing. These events are generally associated with an extensive and variable obstruction of the lung airflow, often irreversible with or without treatment [1].

In the last four decades, there has been a sharp increase in the global prevalence, morbidity, and mortality associated with asthma, and it has calculated that approximately 300-million people worldwide undergo asthma, having world prevalence between 1 and 18%. In México, between 5 to 15% of people suffer it [2], affecting in greater proportion adult women and men younger than 14 years. Around 250 patients/year die, especially > 45 years old [3].

Different extrinsic and intrinsic factors are recognized and, even when it is a common chronic illness, asthma etiology is at day unknown. It has described that the interaction between genetic background and environmental factors, as the pollution and tobacco smoke in intrauterine life and during childhood [4,5], have a determinant role in the development and suffering of the illness. However, the genetic component remains as one of the least investigated factors, even once it could explain the susceptibility of the disease.

Several association studies in diverse populations have reported a high number of different *loci* and genes associated with asthma and clinical phenotypes; for example, *ADORA1* [6], *hCLCA1* [7,8], *CTLA4* [9], and *TNF* [10,11,12]. Besides these genes, other protein products have been included in other studies as CCR1 [13], IL-8 [14], IL-6 [15,16], eNOS [17] and FcR1E [18] due to their relevant function in inflammatory processes as the production, differentiation, and maturation of different cell types, mucous secretion, and inflammatory markers release.

Other closely related genes to this inflammatory process are *IL4* and *IL3*, previously described in genetic susceptibility with asthma [19,20,21]. Furthermore, these cytokines are related to the recruitment of polymorphonuclear cells, for example, basophils and eosinophils and increasing IgE levels [18,22,23,24].

Recent investigations have shown IL-4 increased in serum, bronchoalveolar lavage fluid (BALF), and sputum in asthma patients and animal models, as well as different asthma phenotypes, as rhinitis, severity or atopy [19,25,26]. The increased IL-4 levels are associated with monocytes to M2 type macrophages polarization, increased count of Th17 and Th2 cells, inflammasome activation, impaired expression of immunoglobulin A (IgA) and B-cell autophagy [19,25,27,28,29].

On the other hand, IL-13 has been associated with polarization to Th2 immune response, promoting the resistance to corticosteroids and the production of chemokines as CCL13 [30], and increasing the IgE production [31]. Quantification of IL-13 in mild to severe asthma patients reveals high serum concentration that promotes changes in cellularity, IgE concentration, and decreased lung function (FEV1) [32,33].

A meta-analysis has revealed the association of different single nucleotide polymorphisms (SNPs) in the IL4 gene associated with susceptibility to asthma in Caucasian-European populations, especially the rs2070874 (C-33T) and rs2243250 (C-589T), located in the promotor region [34]. Interestingly, rs2070874 is one of the most reported SNP associated with the risk of suffering asthma, especially in Caucasian and Asian populations, related principally to the severity of the illness [35,36,37,38]. Meta-analysis has also been applied in the IL13 gene to elucidate the risk implied in some SNPs. The most common variants associated with higher risk to suffer asthma were rs2044, rs1295686, and rs1800925, with higher risk to suffer asthma [39,40,41]. Replication in other populations have reported similar findings with the SNPs mentioned previously, mainly in European, Chinese and Middle East cohorts/populations, and with characteristic as the predominance of eosinophils, increased levels of IgE, atopy, higher risk combined with environmental factors and severe case in children [42,43,44].

Other studies have explored the copy-number variations (CNVs) and variable number tandem repeats (VNTR) [45] as possible risk factors to suffer asthma; however, the studies are limited and without populational replications that can corroborate the findings. Also, some investigations have planted the importance of the interaction between genetic and environmental factors, finding increased risk when both are present [46], especially in tobacco smoking or secondhand exposition, in the case of children [47,48,49].

Based on all mentioned above, we aimed to describe the effect of genetic variants type SNPs and CNVs in *IL4* (rs2070874 C/T, nsv528281 and nsv529021), and *IL13* (rs1800925 C/T and rs20541 G/A) genes in the asthma susceptibility, and the possible effect in the serum levels of IL-4, IL-13, IgE and the potential effects implied in the risk environmental exposure factors.

## 2. Materials and Methods

### 2.1. Study Population

In this case-control study, we included 492 Mexican-mestizo subjects over 18 years-old divided into two groups; 147 asthma patients (AP) and 345 clinically healthy subjects (CHS). Patients were recruited from clinical services from Instituto Nacional de Enfermedades Respiratorias Ismael Cosío Villegas (INER), with a previous asthma diagnosis. The diagnosis was based on clinical history, physical examination, spirometry data, as well as the criteria established by the American Thoracic Society (ATS); also, no history of respiratory viral or bacterial infections in the last three months. Subjects with other respiratory diseases were excluded. In the control group were included healthy subjects from blood bank service from INER, without asthma family background, and neither food nor drug allergies.

Demographical and exposure to risk factors data were collected by a personal questionnaire. Data on arterial blood gas measurements, routine blood markers, blood cellularity, lung function, and pharmacological treatment were obtained from clinical records. Cases were classified by severity based on the Global Initiative for Asthma (GINA) 2006. The Strengthening the Reporting of Genetic Association Studies (STREGA) guidelines were taken into account in the design of this genetic association study [50].

### 2.2. Biologic Samples

All participants fulfilled a hereditary-pathology background survey. Exclusion criteria included being of non-Mexican ancestry and having chronic respiratory diseases other than asthma and/or inflammatory disorders. After being given a detailed description of the study, patients who met the inclusion criteria were invited to participate. All participants signed a written informed consent and were provided with a privacy statement describing the protection of personal data. The research protocol was reviewed and approved by the Ethics in Research Committee of the INER in Mexico City (approbation code: B38-11). After signing the informed consent, a blood sample was drawn by venipuncture of the forearm of each patient. From each sample, we proceed to separate serum and plasma for storage at −80 °C until use.

### 2.3. DNA Extraction

The DNA extraction was carried out from EDTA tubes from the plasma pellet with the commercial BDtract DNA isolation kit (Maxim Biotech, San Francisco, CA, USA) and then rehydrated in TE buffer (Ambion, Waltham, MA, USA). Subsequently, the extracted material was quantified using a Nanodrop 2000 system (Thermo Scientific, Wilmington, DE, USA) and stored at −80 °C until use.

### 2.4. Genotyping of CNVs and SNPs

The SNPs were chosen by bibliography research. Those variants reporting a significant association, gene location, possible post-translational changes in the protein, and minor allele frequency (MAF) in Mexican from the Los Angeles population (Mex-LA), reported in the HapMap database (www.hapmap.org) were included. For CNVs selection, we consult the database of genetic variants (DGV) to localize two CNVs structural variants in the *IL4* gene; one of these variants is a deletion between exon 3 and 4. SNPs and CNVs molecular information is presented in Table 1.

SNPs genotyping was performed by real-time PCR (qPCR) and commercial TaqMan DNA probes (Applied Biosystems, Carlsbad, CA, USA). Briefly, a DNA template, a probe sequence-specific for each SNP, and TaqMan Universal PCR Master Mix (Applied Biosystems; Woolston, UK) arranged into 96-well microplates. Besides, three wells without a template included for each genotyping plate (contamination controls). For ~5% of the samples included in the study, genotyping in duplicate for control allele assignment was done.

TaqMan specific sequence probes (Applied Biosystems, Carlsbad, CA, USA) were selected for each CNVs (Table 2). The reaction included DNA template, probe sequence-specific for CNV, and TaqMan Universal PCR Master Mix (Applied Biosystems; Woolston, UK) that were placed into 96-well microplates. Since this is the first time nsv528281 and nsv529021 are evaluated in a Mexican mestizo population, we decided to include subjects with ancestry different from the Mexican (*n* = 15) and subjects with Amerindian ancestry *n* = 10 (five Otomíes and five Mazahuas). These groups were only used in the genotyping of the CNVs as a population control group, looking for samples with probable different copy-number in the analyzed variants.

### 2.5. Plasma Cytokines Measurements

The IL-4 and IL-13 measurements were made in de plasma samples through Human IL-4 and Human IL-13 Standard ABTS ELISA Development Kits (Peprotech. Rocky Hill, NJ, USA) and following supplier instructions. The standard curve and sample measurement were carried out by triplicate.

### 2.6. Statistical Analysis

Quality control and Hardy-Weinberg equilibrium (HWE) were carried out using Plink v. 1.0.7 [51] and De Finetti v.3.0.8 [52], respectively. Alleles and genotypes were compared by different genetic models using Epidat, Epi Info v. 3.1 [53], and Plink v. 1.0.7, and clinical variables were analyzed using SPSS v. 24.0 and the correlation between variables calculated with Spearman’s Ro test in R studio [54]. Plots and graphics were obtained using the ggplot package for R studio. CNV analysis was carried out with the software CopyCaller (Applied Biosystems; Woolston, UK) based on quality value Z and ΔΔCT.

## 3. Results

### 3.1. Population Description

In the present study, we included 486 Mexican-mestizo subjects, 141 AP, and 345 CHS. In both groups, we reported a median age of 40 years old; we found a significant difference in men-women proportion, finding a higher proportion of women in the AP group (67.4%), while in the CHS group, men were slightly higher (57.9%). In the exposure to environmental risk factors, the AP group had a higher percentage of exposed to biomass-burning smoke (BBS) than in the CHS group (47.5% vs. 20.9%, *p* < 0.01, OR = 3.4, 95%CI = 2.25–5.22); in the opposite, among subjects in the CHS group, tobacco-smoking in most frequent than in the AP group. We found ~30% of AP group had no pharmacological treatment; the rest of the patients had at least bronchodilator treatment (~69%); 43% of the treated patients had inhaled corticosteroids as therapy, while 9% was oral glucocorticoid. In the clinical laboratory testing, we found differences in the five cellular populations among groups. The leukocyte and neutrophils are increased in the AP group; while, lymphocytes, eosinophils, and basophils are decreased in the same group (Table 2). Arterial blood gas parameters, glucose, urea, and creatinine values were reported within normal parameters (Appendix A).

### 3.2. Allele and Genotypes Comparison

SNPs included in the study accomplished HWE, although rs1800874 has a slight variation in the proportion of heterozygous (*p* = 0.06, Table 3). De Finetti plots were used to demonstrate HWE (Figure 1). The MAF in all SNPs was similar to previously reported in international databases.

We include three SNPs localized within two genes related to immune response; one in *IL4*; rs2070874 (common allele: T, minor allele: C), and two in *IL13*; rs20541 (common allele: G, minor allele: A) and rs1800925 (common allele: C, Minor allele: T).

For rs20541, the G allele frequency reaches 65%, while A is around 35%, considering cases and control. In the case of rs1800925, the C allele frequency is ~69%, and T is about 31%. Once rs2070874 is analyzed, the frequency for common and minor allele frequency are similar, while in the AP group, the C variant is the common allele (50%), but in the CHS group, T is the common allele (52%). No significant differences were found in the AP vs. CHS comparison. (Table 4).

Next, we calculate the genotype frequencies for each SNPs. In the rs20541, the homozygous for the common allele (GG) was found in around ~42%, heterozygous (GA) in approximately 46%, and homozygous to minor allele (AA) was almost 35%, including cases and controls; to note, GA is the most common genotype in our population. For the rs2070874, the most frequent genotype is the heterozygous (TC, ~48%), followed by CC (~27%) and at last TT (~25%). For the last SNP, rs1800925, the most frequent genotype is heterozygous CC (around 48%), next CT (~39%), and in the final, TT (~11%). After all, we compare genotype frequencies applying the co-dominant model, but no significant association was found for any SNP (Table 4).

Besides, in those asthma patients who have a complete medical record, a stratified analysis based on phenotypes associated with asthma, as rhinitis, allergies, and cellularity, was performed. No significant associations were found for any of the stratified association groups. Data are presented in Appendix A.

### 3.3. Association Analysis Applying Genetics Models

We decided to apply different genetic models to compare genotyping frequencies; results for full-genotype, dominant, and recessive models for each SNP are described in Table 5. No significant association was found for any SNP.

### 3.4. Linkage Disequilibrium (LD) and Haplotype Block (HB) Construction

Since the three SNPs were localized in the same chromosome, LD and HB were calculated. No significant LD or HB was found in any possible combination for the SNPs (Figure 2A). We include the frequencies for all the possible permutation with all the SNPs (Figure 2B).

### 3.5. Copy-Number Variation: Identification and Frequency Comparison

In all included subjects for case-control comparison, was detected the common allele (one-copy per chromosome), no alleles harboring copy-numbers different to the common allele were found. The same behavior happens in the subjects included as a populational reference; only the common allele was discovered (Figure 3).

### 3.6. IL-4 and IL-13 Measurement and Analysis

We made an analysis stratified by genotypes in three different models: full-genotype, dominant, and recessive. The IL-4 levels were stratified according to rs2070874 in the full-genotype model, we found an increase in the TC genotype but were no significant when compared with the other genotypes (*p* = 0.15) using Kruskal-Wallis test and after applying the post hoc test. The IL-13 levels were stratified by the rs20541 and rs1800925 in full-genotype model comparing by the Kruskal-Wallis test and post hoc analysis. No differences were found for any SNP by this model (rs20541 *p* = 0.19; rs1800925 *p* = 0.49), but this was used to organizing a second analysis based on dominant (rs20541) and recessive models (rs1800925) and comparing with Mann-Whitney’s test. No significant differences were found for the applied models (Figure 4).

### 3.7. Correlation between BBS and Clinical Data

When exposure variables were analyzed, in the BBS exposure was detected an increased percentage among subjects in the AP group; for that reason, we decided to use Spearman’s Ro correlation and including genotypes to stratify. No significant correlation was found for any clinical variables or SNP (Appendix A, Appendix A).

### 3.8. IgE Levels Analysis

IgE levels were measured in the AP group as part of the clinical protocol, so we decided to stratify by the genotypes of rs20541/IL4. For this analysis, full-genotype and recessive models were applied. No statistical differences were found for any analyzed model (full-genotype, *p* = 0.89; recessive, *p* = 0.68) (Figure 5).

## 4. Discussion

Asthma is a complex disease consequence of the interaction of intrinsic and extrinsic factors; the current study focused on the possible relation between the genetic component and the risk of developing the disease or clinical characteristics in a Mexican mestizo population. We found significant differences in some demographical variables, like sex, weight, and height, but no in the BMI. In the exposure variables, we observed a higher percentage of subjects exposed to BBS in the AP group (47.5%) with considerable exposition time (~13.5 years). The socioeconomic status in the population could explain differences in the exposition factors; unfortunately, we do not count with the corresponding data to adjust for this potential confounder. Most of the participants attend the INER asthma clinic; INER is a third-level hospital; most of the attending people come from different places of the country, especially from rural and suburban zones. Several authors have described the continuous exposure to BBS in these zones and especially in the female and children populations [55,56,57,58]. Currently, more than 3 billion people in the world; mainly in countries with low to middle income as Mexico, use biomass fuel and coal for cooking or heating purposes, and it is projected that the overall use of solid fuels will keep on rising shortly [59]. This phenomenon is well characterized in the poor sectors of the population and, in México, even in suburban regions of the greatest cities [55]. The most common materials used biomass fuels are wood, crop residues, and animal dung. In our study, almost half of the asthma patients described exposure to BBS in a part of life, some of them in childhood or even prenatal stages. Some studies have demonstrated a higher risk associated with suffering or developing asthma in adults exposed to biomass-burning smoke [56,57,60]. Conversely, a lower percentage of patients are current smokers; this could be due for the clinical advice to asthma patients to avoid tobacco-smoking or contaminants to prevent crisis or exacerbation events [61,62]. In Mexico, about 21% of the population between 12 and 65 years-old are active smokers; at the INER [63], chest physicians counseled the asthma patient about the importance of smoking cessation.

Other changes that could be observed are the decrement of eosinophils and the increased percentage of neutrophils. The most studied cells described in asthma are eosinophils [64,65,66], neutrophils [67,68,69], and mast cells [70], which are responsible for regulating the inflammatory response, polarizing it to a Th2 profile, as well as increasing IgE levels [71,72]. The cellular predominance has helped to identify specific phenotypes of the disease; neutrophilic phenotype stands out here, which has been associated with severe cases of the disease [72,73]. The changes in eosinophils percentage could be due to the treatment with glucocorticoids/corticosteroids; almost half of the AP group is treated with inhaled corticosteroids or oral glucocorticoids. The glucocorticoids affect the levels of Th2 interleukins directly; IL-2, IL-4, IL-10, and IL-13, lowering the IgE levels, decreasing the percentages of the cellular populations as dendritic, eosinophils and mast cells, and in some cases, improving the lung function [74,75,76,77]. In the case of inhaled corticosteroids, it affects lymphocytes, macrophages, and neutrophils’ degranulation directly [78,79,80]. Evidence of this hypothesis is the lower percentages of lymphocytes, other target cells of the treatment.

In the case of neutrophils, there is no evidence about the direct effect of corticosteroids/glucocorticoids in these cells. On the other hand, neutrophils could be stimulated by environmental pollution or smoking [81,82,83,84]. Recent research into the cellularity that characterizes asthma has allowed proposing the classification and study of the disease based on the dominance of certain strains [72]. It is suggested that the prevalence of neutrophils in asthma is associated with more severe inflammatory phenotypes, mediated by T-cell chemokine attractants [85]. This hypothesis is supported by the detection of extracellular DNA in sputum, one of the main mechanisms of action of neutrophils, and detecting high levels of IL-1β [86,87]. Another phenotype associated with increased neutrophils is resistance to steroid treatment, which promotes the worsening of asthma [71]. Neutrophilia has been explored in elderly asthma patients and severity, finding a higher percentage of neutrophils in elderly asthma patients; furthermore, these patients present severe forms of asthma [88].

Regarding pro-inflammatory cytokines, we did not find significant differences in IL-4 nor IL-13 levels. IL-4 is a cytokine related to the polarization of inflammatory profile to Th2 in lymphocytes; it is produced mainly by eosinophils, basophils, and mast cells [22,89]. IL-13 is produced by T-cells that accomplish function as promote cell growth and B-cells differentiation [20,21]. In some studies, the authors had reported increasing in IL-4 and IL-13, but mainly correlating with the presence of neutrophils and mast cells [19,90].

For the genetic association study, we did not find significant results with the illness or any clinical phenotype. Our report is the first in the Mexican population, including these variants; for that reason, we conducted a complete analysis stratifying the genotypes by co-dominant, full-genotype, recessive, and dominant models, but with any of the models found significant associations. Possibly due it is necessary to increase the sample size; when we compare the highest vs. normal levels of leukocytes, we found a trend with the TT genotype of rs1800925 (*p* = 0.08; OR = 0.4, CI = 0.12-1.33). In a previous report in the Asian population, the rs2070874 was associated with the risk of severe forms of asthma [29,37]. These findings have been replicated in a Caucasian population with severity [36], but the positive results are scanty. In the case of rs1800925 and rs20541, the main finding in Asian and some Caucasian populations is with asthma severity and changes in the serum levels of IL-13 [37,39]. We explore the possible association with clinical data, blood markers, and arterial gas measures, but no association was found.

Following a similar procedure in the analysis of the continuous clinical variables (IL-4, IL-13, and IgE), we applied the models above described, but no significant associations or correlations were found. Also, in the correlation analysis, a bordering correlation between years of exposition and neutrophils (%) with the CC genotype of rs1800925 (*p* = 0.055) was found, but prospective studies with a most robust sample size are needed to confirm this hypothesis.

For CNVs, we did not find variation in this kind of polymorphism for any of the two included variants. To the best of our knowledge, there are no studies related to these two CNVs. The inclusion of CNVs aimed to enrich the asthma study with different polymorphisms that could explain the disease complexity.

Among the potential limitations that must be contemplated are the variability of phenotypes, and the lack of adjustment by treatment, as well as not including biological samples that can reflect better the lung environment, as sputum.

In the current study, no association was found between the SNPs/CNVs studied, the disease susceptibility, nor in the plasma levels of the clinical markers. Association studies with a larger sample size are necessary, considering other inflammatory markers related to asthma and clinical phenotypes.

## 5. Conclusions

In the Mexican-mestizo population, SNPs neither CNVs in *IL4* nor *IL13* are associated with asthma susceptibility or involved in plasma cytokine levels. Biomass-burning smoke is a risk factor for asthma susceptibility.

## Figures and Tables

**Figure 1 diagnostics-10-00273-f001:**
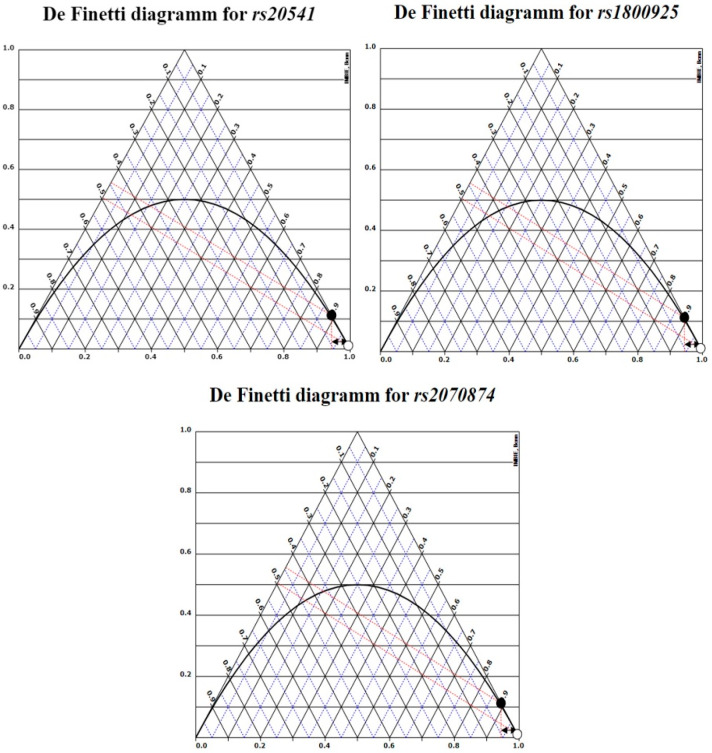
De Finetti plots. Genotypes distribution for each of the SNPs included in the study. White dots are cases (AP) and black controls (CHS).

**Figure 2 diagnostics-10-00273-f002:**
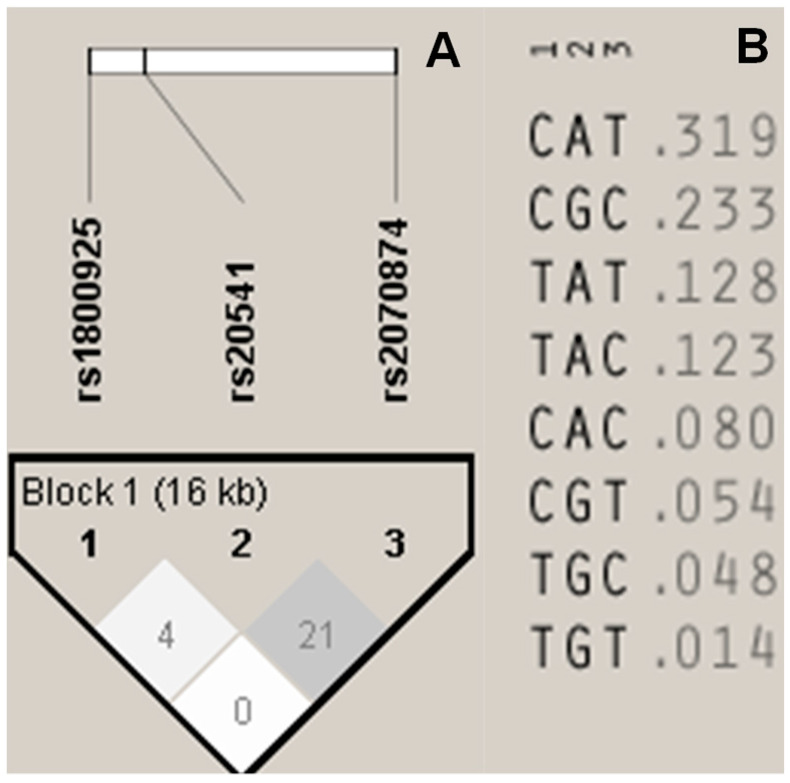
Linkage disequilibrium (LD) analysis. (**A**) Haplotype block, including all the SNPs. Colors and values are shown in R^2^. (**B**) Permutation including each of the alleles for all the SNPs and frequencies.

**Figure 3 diagnostics-10-00273-f003:**
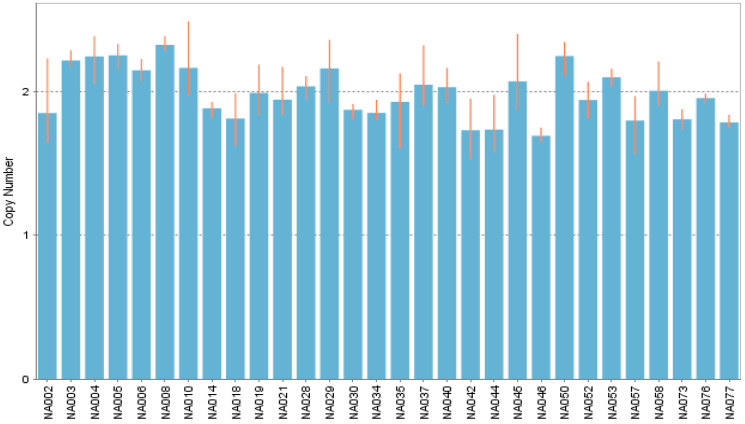
CNV signal analysis by copycaller^®^. The fluorescence signal of each sample is used to calculate the copy number. All of the samples have the common variable (one copy by chromosome). Showing an extract of thirty samples in a bar graph.

**Figure 4 diagnostics-10-00273-f004:**
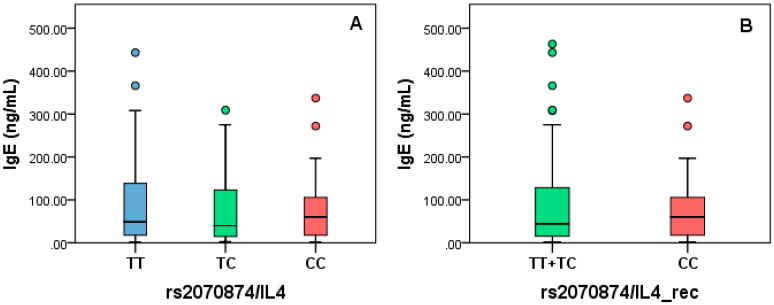
IgE levels stratified by rs2070874 based on full-genotype (**A**) and recessive models (**B**).

**Figure 5 diagnostics-10-00273-f005:**
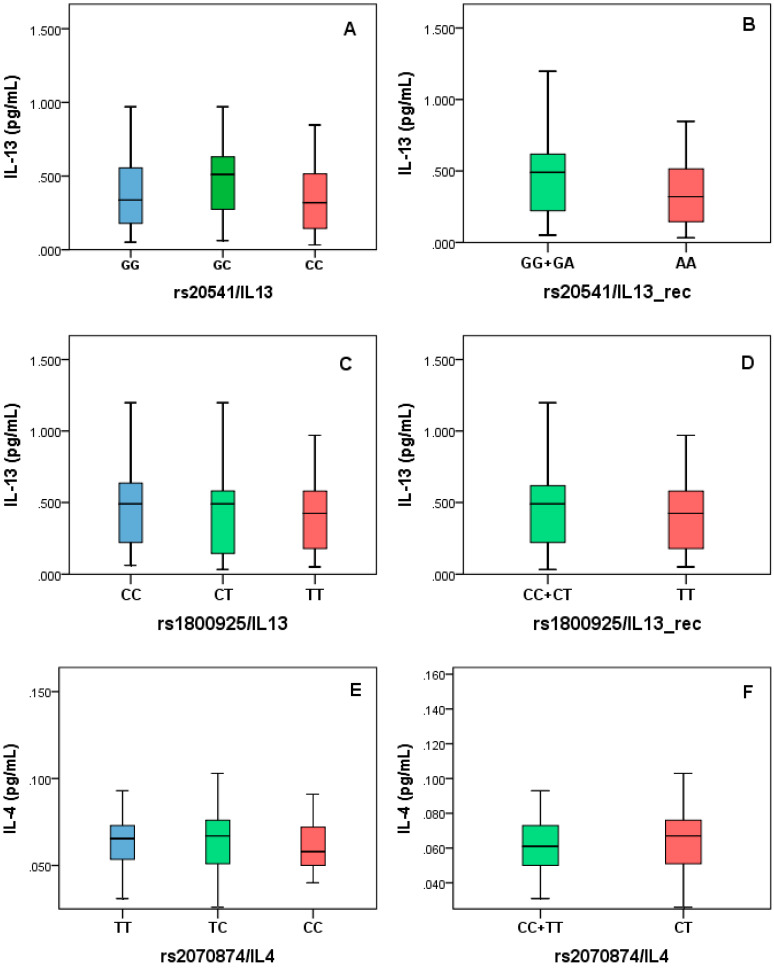
IL-4 and IL-13 levels stratified by genotypes. The full-genotype (**A**,**C**,**E**), recessive (**B**,**D**), and overdominant (**F**) models were applied based on distribution data. No significant differences were identified.

**Table 1 diagnostics-10-00273-t001:** Molecular data for polymorphisms (single-nucleotide polymorphisms (SNPs) and copy-number variants (CNVs)) included.

Gene	SNP/CNV	Protein	Chr	Position (Bp)	Nucleotide Change	Biological Effect	MAF
*IL13*	rs20541	IL-13	5	132,660,272	[G/A]	Missense variant	0.27
	rs1800925			132,657,117	[C/T]	Non-coding transcript	0.25
*IL4*	rs2070874	IL-4		132,674,018	[T/C]	5′UTR variant	0.4
*IL4*	nsv528281	IL-4	5	132,650,771	31,482 bp	Insertion	ND
	nsv529021			132,682,524	67,809 bp	Deletion	ND

MAF: minor allele frequency. Bp: base pairs. Chr: Chromosome. 5′UTR: 5′ untranslated region. ND: Not previously described.

**Table 2 diagnostics-10-00273-t002:** Subjects’ demographical and clinical data.

Variables	AP	CHS	*p*
	*n* = 141	*n* = 345	
**Demographical Data**			
Age (years)	40.0 (29.0–53.0)	40 (30.0–51.5)	0.18
Sex (M/F), %	32.6/67.4	57.9/42.3	<0.01
Weight (Kg)	68.0 (62.0–78.7)	76.8 (68.0–87.2)	<0.01
Height (m)	1.58 (1.52–1.65)	1.64 (1.57–1.70)	<0.01
BMI	27.5 (24.9–31.2)	28.9 (25.7–32.0)	0.08
Pharmacological treatment (%)	97 (68.8%)	NA	NA
Corticosteroids (%)	42 (43.3%)	NA	NA
Glucocorticoids (%)	9 (9.3%)	NA	NA
**Exposure Data**			
Exposed to BBS (%)	67 (47.5)	72 (20.9)	<0.01
Years of exposure to BBS	13.5 (7.5–17.0)	8.0 (5.0–16.5)	<0.01
Smokers (total, %)	68 (48.2)	151 (43.7)	0.42
Current (%)	18 (26.5%)	80 (52.9)	<0.01
Former (%)	50 (73.5%)	71 (47.1)	<0.01
Cigarettes per day	3 (1–10)	2 (1–6)	0.44
Years of smoking	5 (2–11)	5 (2–15)	0.15
Never smoker	73 (51.8)	194 (56.7)	0.27
**Lung Function Data**			
FEV1 * (%)	38.5 (28.5–67.8)	NA	NA
FVC * (%)	54.5 (46.5–134)	NA	NA
FEV1/FVC * (%)	58.2 (49.0–68.8)	NA	NA
FEV1 **(%)	48.0 (42.8–67.0)	NA	NA
FVC ** (%)	74.5 (57.0–85.8)	NA	NA
FEV1/FVC ** (%)	65.5 (56.8–82.5)	NA	NA
Reversibility (%)	19.2 (15.5–25.3)	NA	NA
**Blood Cell Counting and Routine Biomarkers**			
Leucocytes (%)	10.7 (8.5–13.3)	6.7 (5.8–7.6)	<0.01
Neutrophils (%)	84.8 (72.9–91.6)	57.6 (51.9–62.0)	<0.01
Lymphocytes (%)	8.4 (5.6–16.7)	32.1 (27.4–36.4)	<0.01
Eosinophils (%)	0.8 (0.1–2.8)	2.4 (1.6–3.4)	<0.01
Basophils (%)	3.0 (1.0–4.0)	3.0 (2.0–4.0)	0.02
Hemoglobin (g/dL)	15.2 (13.8–16.1)	16.1 (14.8–16.9)	<0.01
MCV	90.7 (87.0–93.3)	89.8 (87.3–92.4)	0.34
MCH	30.0 (29.0–31.0)	30.0 (28.5–31.5)	0.57

Data median and interquartile range (IR 25–75) are shown for all the variables. For the comparisons, Mann-Whitney’s U test was applied for continuous quantitative variables and Fisher exact test for qualitative variables. For significant differences *p*-value < 0.05 was considered. * pre-bronchodilator test. ** post-bronchodilator test. AP: asthma patients; CHS: clinically healthy subjects; FEV1: decreased lung function; NA: Not apply.

**Table 3 diagnostics-10-00273-t003:** Hardy-Weinberg equilibrium.

SNP	Allele	OH	EH	*p*
rs20541	A	0.46	0.46	1.00
rs1800925	T	0.38	0.43	0.06
rs2070874	C	0.53	0.50	0.39

OH: observed heterozygosity. EH: Expected heterozygosity.

**Table 4 diagnostics-10-00273-t004:** Alleles and genotype comparison AP vs. CHS.

Gene/SNP	AP	CHS	*p*	OR	CI 95%
	*n* = 141 (%)	*n* = 345 (%)			
***IL13* rs20541**					
GG	59 (41.84)	145 (42.03)	1.00 (Ref)		
GA	67 (47.52)	157 (45.51)	0.81	1.05	(0.69–1.59)
AA	15 (10.64)	43 (12.46)		0.86	(0.44–1.66)
G	185 (65.60)	447 (64.78)	0.82	1.04	(0.78–1.39)
A	97 (34.40)	243 (35.22)		0.97	(0.72–1.29)
***IL13* rs1800925**					
CC	66 (46.81)	155 (50.72)	1.00 (Ref)		
CT	60 (42.55)	149 (37.39)	0.64	0.95	(0.62–1.43)
TT	15 (10.64)	41 (11.88)		0.86	(0.45–1.66)
C	192 (68.09)	459 (69.42)	0.94	0.98	(0.73–1.32)
T	90 (31.91)	211 (30.58)		1.02	(0.76–1.37)
***IL4* rs2070874**					
TT	36 (25.5)	90 (26.09)	1.00 (Ref)		
TC	68 (48.66)	181 (52.46)	0.44	0.94	(0.58–1.51)
CC	37 (26.24)	74 (21.44)		1.25	(0.72–2.17)
T	140 (49.64)	361 (52.32)	0.48	0.89	(0.68–1.19
C	142 (50.35)	329 (47.68)		1.11	(0.84–1.47)

Allele and genotype frequencies comparison by Fisher’s exact test. For statistical significance was considered a *p* < 0.05. Ref: Genotype employed as a reference for statistical analysis. CI: Confidence intervals. OR: Odds ratio. AP: Asthma patients group. CHS: Clinically healthy subjects group.

**Table 5 diagnostics-10-00273-t005:** Genotypes comparison for AP vs. CHS applying different models.

Model	Genotype	AP	CHS	*p*	OR	CI 95%
		*n* = 141 (%)	*n* = 345 (%)			
***IL13* rs20541**						
**Full-Genotype**	GG	59 (41.84)	145 (42.03)	1.00	0.99	(0.67–1.47)
	GA	67 (47.52)	157 (45.51)	0.69	1.08	(0.73–1.61)
	AA	15 (10.64)	43 (12.46)	0.65	0.84	(0.45–1.56)
**Dominant**	GG	59 (41.84)	145 (42.03)	1.00	0.99	(0.67–1.48)
	GA + AA	82 (58.16)	200 (57.97)		1.01	(0.68-1.49)
**Recessive**	GG + GA	126 (89.63)	302 (87.54)	0.64	1.12	(0.64–2.23)
	AA	15 (10.64)	43 (12.46)		0.84	(0.45–1.56)
***IL13* rs1800925**						
**Full-Genotype**	CC	66 (46.81)	155 (50.72)	0.76	1.08	(0.73–1.59)
	CT	60 (42.55)	149 (37.39)	0.92	0.97	(0.66–1.45)
	TT	15 (10.64)	41 (11.88)	0.76	0.88	(0.47–1.65)
**Dominant**	CC	66 (46.81)	155 (50.72)	0.76	1.08	(0.73–1.59)
	CT + TT	75 (53.19)	190 (49.27)		0.93	(0.63–1.73)
**Recessive**	CC + CT	126 (89.36)	304 (88.11)	0.75	1.13	(0.61–2.12)
	TT	15 (10.64)	41 (11.88)		0.88	(0.47–1.65)
***IL4* rs2070874**						
**Full-Genotype**	TT	36 (25.50)	90 (26.09)	1.00	0.97	(0.62–1.52)
	TC	68 (48.66)	181 (52.46)	0.42	0.84	(0.57–1.25)
	CC	37 (26.24)	74 (21.44)	0.28	1.30	(0.83-2.05)
**Dominant**	TT	36 (25.5)	90 (26.09)	1.00	0.97	(0.62–1.52)
	TC + CC	105 (74.5)	255 (73.91)		1.03	(0.66–1.61)
**Recessive**	TT + TC	104 (73.76)	271 (78.56)	0.28	0.77	(0.49–1.21)
	CC	37 (26.24)	74 (21.44)		1.30	(0.87–2.05)

Allele and genotype frequencies comparison by genetic models. For statistical significance was considered a *p* < 0.05. CI: Confidence intervals. OR: Odds ratio. AP: Asthma patients group. CHS: Clinically healthy subjects group.

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
