# Peer review of "Single Nucleotide and Copy-Number Variants in IL4 and IL13 Are Not Associated with Asthma Susceptibility or Inflammatory Markers: A Case-Control Study in a Mexican-Mestizo Population"

_diagnostics, 2020, doi:10.3390/diagnostics10050273_

Round 1

Reviewer 1 Report

The manuscript entitled Single nucleotide and copy-number variants in IL4 and IL13 are not associated with asthma susceptibility or inflammatory markers: A case-control study in a Mexican-mestizo population is an interesting, however negative study  which shoved no association between IL4 or IL13 SNP and asthma susceptibility. The study contains some flaws:

  1. The asthma patients was characterised by lower eosinophil number compared to controls. Why? The information about corticosteroid treatment (inhaled and oral) should be included in Table 2
  2. The manuscript lacks any methodological data of biomass burning exposure measurements but is correlated with results and discussed although it is not related to the aim of the study. Please upgrade this section in material and methods.
  3. Asthma is very heterogenous disease with many phenotypes. The authors correlated the blood cellularity and evaluated SNPs - the negative results are presented in supplementary file. Maybe the authors should try to perform analysis in well characterised asthma groups. Please compare the prevalence of evaluated SNP in asthma group stratified according to blood eosinophilia (low/high), atopy status, early- late-onset asthma, frequency of exacerbations, dose of used corticosteroids, low or high biomass burning time of exposure e.t.c.

Author Response

  1. The asthma patients was characterised by lower eosinophil number compared to controls. Why? The information about corticosteroid treatment (inhaled and oral) should be included in Table 2

A1: Before starting, let me thank you the observations made to our manuscript. The AP group was formed by asthma stable patients with pharmacological treatment that could decreased the percentage of cells related to Th2 immunologic response, as eosinophils. Now, information about treatment is included in Table 2.

  1. The manuscript lacks any methodological data of biomass burning exposure measurements but is correlated with results and discussed although it is not related to the aim of the study. Please upgrade this section in material and methods.

A2: Thanks for the observation. All exposition data were self-reported by a personal questionnaire. We have expanded the information in the material and methods section.

  1. Asthma is very heterogenous disease with many phenotypes. The authors correlated the blood cellularity and evaluated SNPs - the negative results are presented in supplementary file. Maybe the authors should try to perform analysis in well characterised asthma groups. Please compare the prevalence of evaluated SNP in asthma group stratified according to blood eosinophilia (low/high), atopy status, early- late-onset asthma, frequency of exacerbations, dose of used corticosteroids, low or high biomass burning time of exposure e.t.c.

A3: We really appreciate your comments. In the current version we have addressed your requests. The stratified analysis based on different clinical characteristics can be located in the supplementary table 2 and 3.

Reviewer 2 Report

Please review and extend the introduction section with  few data regarding IL4 and IL13 gene.

The conclusion section have to be improved with the latest studies in this field.

Author Response

  1. Please review and extend the introduction section with few data regarding IL4 and IL13 gene.

A1: First, let me thank you for the time dedicated to reviewing our manuscript, we really appreciate it. We have improved the data, function, and findings related to IL-4 and IL-13 in asthma in the introduction section.

  1. The conclusion section have to be improved with the latest studies in this field.

We have improved the redaction of discussion and conclusion based on relevant literature.

Reviewer 3 Report

Ms. Diagnostics 779866

General comments:

This manuscript studies Mexican, Mestizo or mainly Mestizo, patients with asthma compared to healthy controls with regards to SNPs and potential copy number variants (CNVs) in the IL4 and IL33 genes, and their possible associations with asthma, features of asthma, and cytokine levels. No statistically significant associations regarding these factors were found. The one significant finding was that exposure to biomass-burning smoke was higher in the asthmatic group than in the normal group.

Major comments:

  1. Figure 4: Please show some measure of variation, if the distributions are normal, then standard deviation; if not, for instance, median and quartiles.
  2. As the biomass-burning exposure is the significant result, please expand the text on what is known from other studies in the world on such exposure and asthma.

Minor comment:

3. Table 2: Why may it be that the AP group has lower weight and height than the CHS group?

4. Table 2: The percentage of neutrophils in AP seems very high. Why may this be? May patients in the AP group have ongoing infections?

5. Table 2: The eosinophil percentage, on the other hand, in AP appears low. What is the possible reason for this? Are some patients in AP on drugs that are expected to decrease eosinophils?

6. Do you have any lung function data on AP?

7. Figure 1: The font of the numbers and text in the graphs is very small, too small to read at 100% on the screen or if printed out. Please enlarge.

8. Fig. 4: Also here the numbers, particularly the concentrations on the y axes, are difficult or impossible to see. Please make larger.

9. Supplementary table 1: Do you have these data for CHS?

10. Suppl. Fig. 1: Are the light blue areas confidence intervals? Please indicate and if so what percentage of confidence.

11. Please provide legend text covering the graphs below Suppl. fig. 1 legend.

12. Introduction: Are there really only 290,000 persons suffering from asthma in Mexico? That sounds surprisingly low. If the prevalence were 1%, there should be more than one million. Please discuss. Is asthma likely severely underdiagnosed in Mexico?

13. Line 96: Should it be “Los Angeles” (and not “Angeles”)?

14. Lines 113-114: Were the Otomis and Mazahuas included in the main group or analyzed separately? If analyzed separately, were the results different from the Mestizos?

15. What were the detection limits in the ELISAs?

16. Statistical analysis: Were all data normally distributed? If not, please use Spearman correlation instead of Pearson.

17. How was exposure to biomass-burning smoke determined or reported?

18. The manuscript needs linguistic revision. For example on line 39 it says “it has calculated”, should be “it has been calculated”.

Author Response

  1. Figure 4: Please show some measure of variation, if the distributions are normal, then standard deviation; if not, for instance, median and quartiles.

Thanks for your recommendation. To make appropriate images, we change the type of graphics, and now we are showing box plots.

  1. As the biomass-burning exposure is the significant result, please expand the text on what is known from other studies in the world on such exposure and asthma.

We have added a couple of paragraphs discussing the relevance of biomass-burning smoke exposure and its context in the world and socioeconomic status.

  1. Table 2: Why may it be that the AP group has lower weight and height than the CHS group?

Thank you for your comment. We did not find a plausible explanation of this phenomenon. Partially could be for the prevalence of overweight in the Mexican population. If well we did not include subjects with diabetes, we did not control the weight. Also, subjects in the control group are selected as healthy from the blood donor bank.

  1. Table 2: The percentage of neutrophils in AP seems very high. Why may this be? May patients in the AP group have ongoing infections?

As is stated in the inclusion criteria, participants having infections (in the last three months) and other lung diseases were excluded; now, we have included a couple of paragraphs explaining the cases and controls inclusion criteria and parameters. Also, the discussion includes some articles explaining this based on the literature.

  1. Table 2: The eosinophil percentage, on the other hand, in AP appears low. What is the possible reason for this? Are some patients in AP on drugs that are expected to decrease eosinophils?

Thanks for the observation. We have included a possible explanation about this in the discussion section.

  1. Do you have any lung function data on AP?

Yes. This data have been updated in the clinical data, table 2.

  1. Figure 1: The font of the numbers and text in the graphs is very small, too small to read at 100% on the screen or if printed out. Please enlarge.

Thanks for the comment. The images' titles were enlarged.

  1. 4: Also here the numbers, particularly the concentrations on the y axes, are difficult or impossible to see. Please make larger.

Thanks for the comments. Scales were enlarged.

  1. Supplementary table 1: Do you have these data for CHS?

No. We measured those clinical variables only in cases; due are collected from clinical records.

  1. Fig. 1: Are the light blue areas confidence intervals? Please indicate and if so what percentage of confidence.

You are right; the blue areas are standard error (95%). Now is described in the corresponding figure legend.

  1. Please provide legend text covering the graphs below Suppl. fig. 1 legend.

Thank you, is done.

  1. Introduction: Are there really only 290,000 persons suffering from asthma in Mexico? That sounds surprisingly low. If the prevalence were 1%, there should be more than one million. Please discuss. Is asthma likely severely underdiagnosed in Mexico?

We have corrected the data; but, you are right, as in many middle-income countries, asthma and other respiratory diseases are underdiagnosed.

  1. Line 96: Should it be "Los Angeles" (and not "Angeles")?

Thanks, we have corrected this mistake.

  1. Lines 113-114: Were the Otomis and Mazahuas included in the main group or analyzed separately? If analyzed separately, were the results different from the Mestizos?

Population control groups only were included in the CNVs genotyping. As we explained in the results section, we do not show data of CNV alleles, because, for all populations, we only found the common variants. Now we have improved the redaction trying to clarify this point in the "Material and methods" section.

  1. What were the detection limits in the ELISAs?

From 15.6 to 1,000 pg/mL for IL-4 with a sensitivity <1.5 pg/mL.

From 10.0 to 1,000 pg/mL for IL-13 with a sensivity <10 pg/mL.

  1. Statistical analysis: Were all data normally distributed? If not, please use Spearman correlation instead of Pearson.

Thank you for your crucial observation. Now we have applied Spearman's Rho to analyze the correlation data.

  1. How was exposure to biomass-burning smoke determined or reported?

Environmental exposure data was self-reported by questionary, and we considered hours of exposition and how many years the subject was exposed.

  1. The manuscript needs linguistic revision. For example on line 39 it says "it has calculated", should be "it has been calculated".

Thank you, now the whole manuscript was reviewed for grammar.

Reviewer 4 Report

Thank you for giving me the opportunity to review the manuscript diagnostics-779866, with the title "Single Nucleotide and Copy-Number Variants in IL4 and IL13 Are Not Associated with Asthma Susceptibility or Inflammation Markers: A Case- Control Study in a Mexican-Mestizo Population ". This is a case-control study aims to evaluate the effect SNPs and CNVs in IL4, and IL13  genes in the asthma susceptibility, and the possible effect in the serum levels of IL-4, IL-13, and IgE. The paper has been well written and provides important messages about genotype in asthma. I have some points that the authors need to clarify as following.

  • The clinical phenotypes or populations choose to be enrolled into SNP and CNV studies will great influence the interpretation of the results. In the current study, as you shown in table 2, although the percentage of active smoker were significantly less in the AP group (AP & CHS: 26.5 % & 52.9%), the overall smoking status remains high in each group (AP & CHS: 48.2 & 43.7).
  1. How about the incidence of smoking in Mexico City or in Mexico?

As we know smoking is one of the most important risk factors contribute to COPD, and most patient with asthma has normal lung function test and COPD patients will have abnormal lung function test, the author was suggested to provide the lung function test results (if possible).

  1. How to identify biomass-burning smoke (BBS) in the current study? Are those populations lived in different city or significantly different in social economic status? How to explain the great difference between the two group? (AP & CHS)

  1. The population of asthma enrolled into the current study is “not” the traditional eosinophilic asthma due to high prevalence of smoking status and BSS exposure, therefore the SNP and CNV results were independent from IL-4, IL-13 and IgE expression. (That is because IL-4, IL-13 is correlated to Eosinophilic pathway, and IgE is correlated to mast cell activation in asthma). In addition, from the CBC/DC data in table 2, the AP group was significantly higher in Neutrophil %, compared to CHS group. The eosinophil % in AP group was very low and the value was only 0.8%.

  1. Suggest adds some article to address the issue regarding “Neutrophilic asthma”.

Author Response

  1. The clinical phenotypes or populations choose to be enrolled into SNP and CNV studies will great influence the interpretation of the results. In the current study, as you shown in table 2, although the percentage of active smoker were significantly less in the AP group (AP & CHS: 26.5 % & 52.9%), the overall smoking status remains high in each group (AP & CHS: 48.2 & 43.7).

We appreciate the time, correction, and comments of our manuscript. We hope to answers and complete all the requested information.

The new manuscript version includes information (discussion section) about this phenomenon in Mexico. About 21% of the population between 12 and 65 years-old are active smokers; chest physicians counseled the asthma patient about the importance of smoking cessation; a significant part of the patients follows the recommendations and this is reflected in the current smokers status (26.5%) and former smokers (~74%). In conclusion, the change in percentages could be the effect of medical advice.

  1. How about the incidence of smoking in Mexico City or in Mexico? As we know smoking is one of the most important risk factors contribute to COPD, and most patient with asthma has normal lung function test and COPD patients will have abnormal lung function test, the author was suggested to provide the lung function test results (if possible).

Thanks for the note. We have added the lung function section in table 2 (Demographic and clinical data)

  1. How to identify biomass-burning smoke (BBS) in the current study?

Environmental exposure data was self-reported by questionary, and we considered hours of exposition and how many years the subject was exposed.

  1. Are those populations lived in different city or significantly different in social economic status?

As we stated from the initial submission, probably one of the principal variables affecting is the socioeconomic status. The INER is a reference hospital (third-level), so people from different regions are attended. In the literature, the exposure to BBS is predominant of low to middle-income countries; Mexico is one of the cases; even in suburban regions of Mexico City, the capital of the country, there is a large percentage of people exposed to biomass. It is important to mark that in some cases, the antecedent of BBS exposition was in childhood. In our workgroup, we have described this fact (PMID: 29872290) previously.

  1. How to explain the great difference between the two group? (AP & CHS) The population of asthma enrolled into the current study is "not" the traditional eosinophilic asthma due to high prevalence of smoking status and BSS exposure, therefore the SNP and CNV results were independent from IL-4, IL-13 and IgE expression. (That is because IL-4, IL-13 is correlated to Eosinophilic pathway, and IgE is correlated to mast cell activation in asthma). In addition, from the CBC/DC data in table 2, the AP group was significantly higher in Neutrophil %, compared to CHS group. The eosinophil % in AP group was very low and the value was only 0.8%.

Thank you for your valuable comment again. We agree with you regarding the “no classical asthma” We think that part of the differences could be the effect of pharmacological treatment attenuating the inflammatory response, even including adherence and compliance with treatment could be a factor to consider. Second, a significant part of the CHS group is active smokers; in the literature is described increased inflammatory response in current smokers and even inflammation can last for years still if stimulation ceases (stop smoking).

  1. Suggest adds some article to address the issue regarding "Neutrophilic asthma".

Attending your suggestion, we have added some articles focused on neutrophilic asthma and the consequences.

Round 2

Reviewer 1 Report

The authors have thoroughly addressed my comments and the additional data (Table 2 and Table 3 in Supplementary file) has improved the manuscript.

Minor remarks:

  1. Please correct the numeration of tables in Supplementary file.
  2. Important factor associated with neutrophilic asthma is age – neutrophilic asthma is frequently related to late-onset and elderly age, please add this information into the discussion.

Author Response

  1. Please correct the numeration of tables in Supplementary file.

I want to thank you for the time spent reviewing our manuscript. We corrected the numeration of the tables in the supplementary file.

  1. Important factor associated with neutrophilic asthma is age – neutrophilic asthma is frequently related to late-onset and elderly age, please add this information into the discussion.

Attending your request, We added a short paragraph explaining previous studies related to neutrophilic asthma, severity, and age of patients.

Reviewer 3 Report

General comments:

This revised manuscript has addressed previous comments.

Author Response

No comments for correction.

I want to thank you for the time spent reviewing our manuscript.